# ACTIVE MODEL SELECTION FOR LARGE LANGUAGE MODELS

## ABSTRACT

We introduce LLM SELECTOR, the first framework for active model selection of Large Language Models (LLMs). Unlike prior evaluation and benchmarking approaches that rely on fully annotated datasets, LLM SELECTOR efficiently identifies the best LLM *with limited annotations*. In particular, for any given task, LLM SELECTOR adaptively selects a small set of queries to annotate that are most informative about the best model for the task. To further reduce annotation cost, we leverage a judge-based oracle annotation model. Through extensive experiments on 6 benchmarks with 151 LLMs, we show that LLM SELECTOR reduces annotation costs by up to 59.62% when selecting the best and near-best LLM for the task.

## 1 INTRODUCTION

How can we select the best Large Language Model (LLM) for a given application or data distribution without retraining? Answering this question has become increasingly difficult as the number of readily available models continues to expand. Recent advances in architectures, training strategies, and access to massive datasets have enabled impressive zero-shot capabilities, allowing LLMs to perform a wide range of tasks without task-specific fine-tuning (Wei et al.; Kojima et al., 2022). As a result, a large and diverse collection of pretrained models differing in architecture, training data, and optimization objectives is now easily accessible through academic repositories and commercial platforms (Hugging Face; OpenAI; Google DeepMind; Anthropic).

This abundance of choice, while offering wide flexibility for deployment, also introduces a fundamental challenge for practitioners as the performance differences across these LLMs can be substantial, particularly when transferring across domains, tasks, or languages (Liang et al., 2023). Although significant efforts have been devoted to the evaluation and benchmarking of LLMs (Liang et al., 2023; Fourrier et al., 2024; OpenCompass, 2023), the rapid expansion of both candidate models and evaluation scenarios makes existing practices increasingly difficult to apply for model selection (Chang et al., 2024). In particular, benchmarks often struggle to keep pace with the fast release cycle of new models or frequently focus on narrow or standardized tasks, which may not adequately capture the requirements of domain-specific applications. A common approach to model selection is to rely on random or heuristically chosen small subsets of annotated data (Polo et al., 2024; Vivek et al., 2024), but such approaches often result in suboptimal use of resources and fail to reliably capture differences across models (Kossen et al., 2021). To address this, several studies have explored active model selection (Karimi et al., 2021; Liang et al., 2020; Okanovic et al., 2025; Gardner et al., 2015; Tahan et al., 2024), where limited annotations of strategically chosen subsets are utilized, but this line of work is largely centered around classification tasks rather than generative settings (Okanovic et al., 2025; Kay et al., 2025; Madani et al., 2012; Karimi et al., 2021; Piratla et al., 2021; Liu et al., 2022; Kassraie et al., 2023; Xia et al., 2024a; Li et al., 2024a;b). Thus, to date, how to reliably identify the best LLM for a specific task and data distribution under limited annotation resources remains an open question.

In this work, we address this problem and ask: Given a pool of queries and a set of candidate LLMs, which examples should be annotated in order to reliably identify the best LLM, both in a model-agnostic and annotation-efficient manner?

**Contributions:** In this paper, we introduce the active model selection problem for LLMs and present LLM SELECTOR, a principled framework for selecting the best LLM under a limited annotation

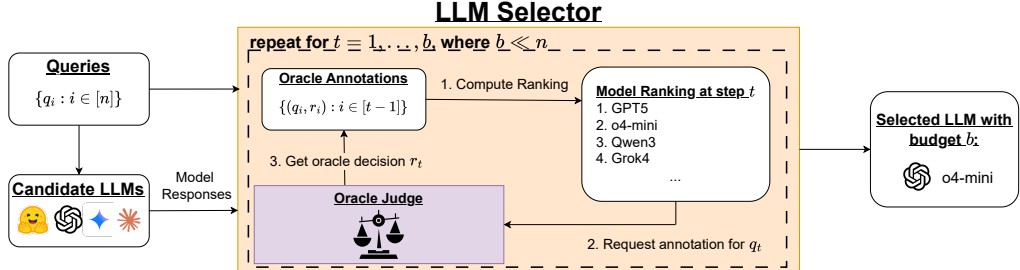

Figure 1: An overview of LLM SELECTOR. For an arbitrary pool of $n$ queries and a set of candidate language models, LLM SELECTOR adaptively annotates most informative $b \ll n$ queries for identifying the best language model for the pool.

budget, along with adapted baseline strategies for this problem. Given a large set of $n$ queries and a limited annotation budget $b$ with $b \ll n$, LLM SELECTOR selects $b$ queries whose annotations are expected to maximally reduce uncertainty about the best model for the entire set. Our approach builds on information-gain criteria (Chen et al., 2015), and quantifies informativeness using a two-parameter model that measures information gain as Shannon's mutual information between the unknown best model and annotations.

Motivated by the growing adoption of judge-based approaches (Zheng et al., 2023; Li et al., 2024c; 2023; Zheng et al., 2023), we employ a judge-based annotation process in which each query is annotated with a vector over candidate models. For each model candidate, we compare its response to the query against that of a baseline model using oracle preference judgments. This judge-based design alleviates the need for costly reference answers or summaries that are known to be far more expensive than pairwise judgments (Zopf, 2018; Ouyang et al., 2022a; Rafailov et al., 2023; Luo et al., 2022; Callison-Burch et al., 2006), and mitigates the noise commonly introduced by reference-based evaluation metrics (Zopf, 2018; Ouyang et al., 2022a; Rafailov et al., 2023; Novikova et al., 2017).

We validate LLM SELECTOR across 6 benchmarks on 151 LLMs. Specifically, our evaluation covers three categories of datasets: (i) *general dialogue*: AlpacaEval (Li et al., 2023), Arena-Hard (Li et al., 2024c), and MT-Bench (Zheng et al., 2023); (ii) *vision-language*: Flickr30k (Young et al., 2014) and Bingo (Cui et al., 2023); and (iii) *medical*: MediQA (Ben Abacha et al., 2019). These benchmarks employ LLM-as-a-Judge for evaluation, which has been shown to correlate strongly with human evaluations, even exceeding the agreement level between human annotators (Zheng et al., 2023). Importantly, our method does not rely on LLM judges specifically and is equally compatible with other oracle judges, such as human evaluators or alternative assessment methods.

LLM SELECTOR shows consistently competitive performance across all experiments, while providing significant reductions in annotation costs on several datasets, with only a small fraction of the annotation budget required by baseline selection strategies where it reduces the annotation costs by up to 58.33%, while achieving a 59.62% reduction when selecting models within a 1% win-rate vicinity of the best model. Moreover, we show that LLM SELECTOR can find near-best models even before exhausting the budget needed to reach the best model, indicating that LLM SELECTOR maintains robust performance under extreme budget constraints.

Once the best LLM is selected based on $b$ annotated queries, we use it to generate outputs for the remaining $n - b$ queries where $n - b \gg b$. Our method is fully model-agnostic: it requires no access to internal parameters and imposes no restrictions on output format, making it directly applicable in black-box or API-only settings. An overview of LLM SELECTOR is shown in Figure 1.

## 2 RELATED WORK

Several methodologies exist for **LLM evaluation**. Traditional multiple-choice (Srivastava et al., 2022; Suzgun et al., 2022), or short-answer benchmarks (Cobbe et al., 2021) provide a standardized way to evaluate model performance, though they do not assess the generative abilities of LLMs. For

tasks such as summarization (See et al., 2017; Narayan et al., 2018) and translation (Goyal et al., 2022), reference-based benchmarks are commonly used, where model outputs are compared against human-written ground truth using metrics like BLEU (Papineni et al., 2002), ROUGE (Lin, 2004), and BERTScore (Zhang et al., 2020). More recently, judge-based evaluation has seen growing adoption. LMArena (Zheng et al., 2023) is a live leaderboard using human annotators. Static benchmarks like Arena-Hard (Li et al., 2024c), AlpacaEval (Li et al., 2023) and MT-Bench (Zheng et al., 2023) rely on LLM-as-a-Judge for automated evaluation. At a higher level, leaderboards such as HELM (Liang et al., 2023), OpenCompass (OpenCompass, 2023), and OpenLLM (Fourrier et al., 2024) aggregate benchmarks measuring models on different capabilities in order to give a full view of LLM capabilities. However, these evaluation methodologies require relying on full access to human annotators or LLM-as-a-Judge, and due to the large scale of modern benchmarks, such evaluations are often not feasible with limited resources.

Most prior work on **active model selection** focus on classification tasks (Zhao et al., 2008; Liang et al., 2020; Gardner et al., 2015; Okanovic et al., 2025; Kay et al., 2025). Some studies consider an online setting, where data arrive sequentially from a stream (Madani et al., 2012; Karimi et al., 2021; Piratla et al., 2021; Liu et al., 2022; Kassraie et al., 2023; Xia et al., 2024a; Li et al., 2024a;b; Xia et al., 2024b). Active model selection has also been studied for LLMs, but limited to scenarios with two candidate models (Tahan et al., 2024) or a single model under active testing (Berrada et al., 2025; Huang et al., 2025). In contrast, our method, LLM SELECTOR, can handle an arbitrary number of candidate LLMs.

Finally, while some prior work explores efficient active ranking based on comparisons (Jamieson & Nowak, 2011; Caron & Doucet, 2012), they primarily select pairs of models for evaluation. By contrast, our setup compares models on LLM queries spanning diverse levels of difficulty, where the outcome of the evaluation depends on the query itself. This motivates a data-centric perspective in which we prioritize selecting examples for annotation rather than model pairs.

## 3 LLM SELECTOR

In this section, we introduce LLM SELECTOR. We first define the problem setting in Section 3.1, and describe our annotation framework based on preference judgments in Section 3.2. In Section 3.3, we present LLM SELECTOR algorithm for annotation-efficient LLM selection. Finally, Section 3.4 details our hyperparameter selection strategy, which requires no oracle annotations.

### 3.1 PROBLEM SETTING

Consider the inference-time scenario in which we are provided with a set of $n$ unannotated queries $Q = \{q_i \in \mathcal{Q} \mid i \in [n]\}$. Each query $q_i$ represents a user-issued prompt or request to an oracle. We denote the oracle-annotated ground-truth response to $q_i$ by $r_i \in \mathcal{R}$. Since these annotations are not observed, we use $R_i$ to denote the unknown response $r_i$.

Given a collection of $m$ pretrained language models $\mathcal{M} = \{f_j : \mathcal{Q} \to \mathcal{R} \mid j \in [m]\}$, our objective is to identify the best language model in $\mathcal{M}$ for producing high-quality responses to the queries $Q$. Because oracle-provided annotations are costly, we assume access to only a limited number of at most $b \ll n$ annotations. The problem therefore reduces to selecting $b$ queries whose annotations provide maximal information about the identity of the best model. We define the best model, denoted by $f^*$, as the model with the highest utility among $M$ if all annotations $\{r_i \mid i \in [n]\}$ were observed. Once identified, we deploy this model to generate responses for the remaining $n - b$ unannotated queries. We define the random variable $F$ to represent the unknown best model.

Formally, we cast the selection problem as one of maximizing mutual information. That is, we aim to identify a subset $\mathcal{A} \subseteq \{(q_i, r_i) \mid i \in [n]\}$ of at most $b$ annotated examples that maximizes the mutual information between $F$ and the selected annotations:

$$\mathcal{A}_{\mathrm{opt}[b]} = \underset{\substack{\mathcal{A} \subseteq \{(q_i, r_i) \mid i \in [n]\} \\ |\mathcal{A}| \leq b}}{\arg\max} \ \mathbb{I}(F; \mathcal{A}). \tag{1}$$

## 3.2 ANNOTATION VIA DIRECT PREFERENCE JUDGMENTS

The evaluation of long-form candidate responses cannot rely on exact string matching, and therefore requires more sophisticated methods. Beyond correctness, factors such as relevance, helpfulness, complexity, and level of detail influence the desirability of an answer. Because reference-based metrics often produce noisy scores (Novikova et al., 2017; Callison-Burch et al., 2006), we instead evaluate models using direct preference judgments (Zheng et al., 2023; Rafailov et al., 2023), which compare model responses pairwise and is shown to be more stable than individual model ratings (Jones et al., 2011; Zopf, 2018). As for LLMs, the preference-based is already being adopted as the evaluation method in many open-ended contemporary LLM benchmarks (Zheng et al., 2023; Li et al., 2024c; 2023).

Formally, for a given query $q_i \in Q$, an *oracle judge* performs a pairwise comparison between the responses of the models $f_j$ and $f_k$. We write $>$, $<$, and $=$ to denote the oracle judge's preference relation, with the following outcomes:

- $f_j(q_i) > f_k(q_i)$: the response of $f_j$ is preferred,
- $f_j(q_i) < f_k(q_i)$: the response of $f_k$ is preferred,
- $f_j(q_i) = f_k(q_i)$: the responses are judged equally good (or equally poor).

We express the pairwise judgment of the oracle as

$$\text{OracleJudge}(q_i, f_j(\cdot), f_k(\cdot)) = \mathbb{1}[f_j(q_i) > f_k(q_i)] + \tfrac{1}{2} \cdot \mathbb{1}[f_j(q_i) = f_k(q_i)],$$

where $\mathbb{1}[\cdot]$ denotes the indicator function.

To compare two models across a collection of queries, we adopt the *win rate* metric (Li et al., 2023). For the query set $Q$, the win rate of $f_j$ over $f_k$ is defined as

$$\text{WR}_Q(f_j, f_k) = \frac{1}{n} \sum_{i=1}^{n} \text{OracleJudge}(q_i, f_j(\cdot), f_k(\cdot))$$

with $\text{WR}_Q(f_j, f_k) + \text{WR}_Q(f_k, f_j) = 1$ for $j, k \in [m]$.

Since identifying the best model through full pairwise ranking requires $\mathcal{O}(m^2)$ oracle annotations, we instead adopt a simplified strategy based on comparisons against a single baseline. Specifically, we designate one of the language models in $\mathcal{M}$ as *baseline model* to reduce the annotation cost further. We denote the baseline model by $\bar{f}$. We select a candidate LLM expected to perform strongly as the baseline, aiming to produce a more informative ranking. We provide details of baseline model selection in Section 4.1. Each remaining model is then evaluated according to its win rate relative to $\bar{f}$, and LLM SELECTOR returns the model the model with the highest win rate based on the annotated queries.

To formally characterize the mutual information between the unknown best model and the annotations, we propose a two-parameter model that describes the behavior of the unknown best language model relative to the baseline, with respect to the oracle preference relation introduced earlier:

$$\begin{aligned}
\mathbb{P}(F(q) < \bar{f}(q)|F = f^*) &= \epsilon_{\text{loss}} \\
\mathbb{P}(F(q) = \bar{f}(q)|F = f^*) &= \epsilon_{\text{draw}} \\
\mathbb{P}(F(q) > \bar{f}(q)|F = f^*) &= 1 - \epsilon_{\text{loss}} - \epsilon_{\text{draw}}
\end{aligned} \tag{2}$$

where $\epsilon_{\text{loss}}, \epsilon_{\text{draw}} \in [0, 1]$ and $\epsilon_{\text{loss}} + \epsilon_{\text{draw}} \leq 1$. The values of $\epsilon_{\text{loss}}$ and $\epsilon_{\text{draw}}$ are determined in advance, following the procedure described in Section 3.4.

## 3.3 THE ALGORITHM

Given the query set $Q$, our objective is to select at most $b$ queries such that, once annotated, they maximize our information about the best language model as defined in Equation 1. To this end, we adopt a sequential information maximization strategy (Chen et al., 2015; Okanovic et al., 2025) for selecting queries one at a time until the budget $b$ is exhausted.

In our sequential framework, let $U_t$ denote the pool of unannotated queries, and $A_t$ the set of annotated queries accumulated up to the sequential step $t$ with $U_0 = Q$ and $A_0 = \emptyset$. At each $t$, we select the next query $q_t$ to annotate as follows:

$$
\begin{aligned}
q_t &= \arg\max_{q \in U_t} \mathbb{I}(F; R \mid A_t, q) \\
&= \arg\max_{q \in U_t} \mathbb{H}(F \mid A_t) - \mathbb{E}_R[\mathbb{H}(F \mid A_t \cup \{(q, R)\})] \\
&= \arg\min_{q \in U_t} \mathbb{E}_R[\mathbb{H}(F \mid A_t \cup \{(q, R)\})],
\end{aligned}
\tag{3}
$$

where $\mathbb{H}(F \mid A_t)$ denotes the conditional entropy of $F$ given the annotations observed up to step $t$.

Selecting the next query reduces to finding the query that minimizes the expected conditional entropy of $F$ given the current annotations as in Equation 3. As oracle responses for unannotated queries are unavailable, we compute this expectation through noisy annotation of the responses to each $q \in \mathcal{U}_t$ using *weak judges*.

### 3.3.1 NOISY ANNOTATIONS VIA WEAK JUDGES

The intuition behind the noisy annotation approach is to evaluate a candidate response by comparing it against the set of possible model responses, assigning higher preference to those that has greater similarity to other candidates.

Formally, we tokenize each response as a sequence of words: $(w_1, w_2, \ldots, w_L)$. For a given $k \in \mathbb{N}$, we construct a language model based on $k$-grams. The estimated probability of a word $w_l$ in this language model is determined by the previous $k-1$ words where $\mathbb{P}(w_l|w_{1:L}) := \mathbb{P}(w_l|w_{l-k+1:l-1})$. We fit the $k$-gram model on the responses of the candidate models $\mathcal{M}$, independently for each query $q$. For computing the average sequence likelihood of a sequence, we average the word probabilities:

$$
\mathbb{P}(w_1, w_2, \ldots, w_L) = \frac{1}{L} \sum_{l=1}^{L} \mathbb{P}(w_l|w_{l-k+1:l-1}).
$$

Comparison of models $f$ and $\bar{f}$ by a weak judge is done by choosing the response with the higher average likelihood, $\mathbb{P}(f_j(q))$ or $\mathbb{P}(\bar{f}(q))$. The weak judge decision with $k$-gram model is denoted as $f(q) >_{(k)} \bar{f}(q)$, $f(q) =_{(k)} \bar{f}(q)$ or $f(q) <_{(k)} \bar{f}(q)$. We use $r_{(k)}$ to represent the noisy annotation made by weak judge $k$. Based on the weak judge decision and parameter model in Equation 2, we can compute the estimated information gain through the following probability:

$$
\begin{aligned}
\mathbb{P}(F = f_j|A_t \cup \{(q, r_{(k)})\}) \propto\ & \epsilon_{\text{loss}}^{\mathbb{1}[f_j(q) <_{(k)} \bar{f}(q)]} \cdot \epsilon_{\text{draw}}^{\mathbb{1}[f_j(q) =_{(k)} \bar{f}(q)]} \\
& \cdot (1 - \epsilon_{\text{loss}} - \epsilon_{\text{draw}})^{\mathbb{1}[f_j(q) >_{(k)} \bar{f}(q)]} \mathbb{P}(F = f_j|A_t)
\end{aligned}
\tag{4}
$$

In total, we have $z \geq 1$ weak judges, each using the $k$-gram model with $k \in [z]$. Given $\mathbb{H}(F|A_t \cup \{(q, r_{(k)})\})$, the estimated entropy by a weak judge, we compute the expected entropy by averaging over all weak judges:

$$
q_t = \arg\min_{q \in \mathcal{U}_t} \frac{1}{z} \sum_{k=1}^{z} \mathbb{H}(F|\mathcal{A} \cup \{(q, r_{(k)})\})
$$

where we use a uniform distribution over weak judges for computing the expectation.

### 3.3.2 UPDATING MODEL POSTERIOR BELIEF

After annotating the selected query at step $t$, we update the posterior belief over the best language model conditioned on all annotations observed up to time $t$:

$$
\mathbb{P}(F = f_j|A_t \cup \{(q, R = r)\}) \propto \mathbb{P}(A_t \cup \{(q, R = r)\}|F = f_j) \cdot \mathbb{P}(F = f_j).
$$

With the two-parameter annotation model in Equation 2, the posterior belief is updated as:

$$\mathbb{P}(F = f_j | A_{t+1}) \propto \mathbb{P}(F(q_t) = r_t | F = f_j) \cdot \mathbb{P}(F = f_j | A_t)$$
$$\propto \epsilon_{\text{loss}}^{\mathbb{1}[f_j(q_t) < \bar{f}(q_t)]} \epsilon_{\text{draw}}^{\mathbb{1}[f_j(q_t) = \bar{f}(q_t)]} (1 - \epsilon_{\text{loss}} - \epsilon_{\text{draw}})^{\mathbb{1}[f_j(q_t) > \bar{f}(q_t)]} \mathbb{P}(F = f_j | A_t)$$

The pseudocode of the algorithm is provided in Algorithm 1.

---

**Algorithm 1** LLM Selector Algorithm

---

**Require:** models $\mathcal{M}$, test queries $\mathcal{Q}$, parameters $\epsilon_{\text{loss}}, \epsilon_{\text{draw}}, \epsilon_3$, labeling budget $b$, number of weak judges $z$, $j \in [m]$
  $\mathcal{A}_0 \leftarrow \{\}, \mathcal{U}_0 \leftarrow \mathcal{Q}$
  `//Uniform model prior`
  $\mathbb{P}(F = f^j | \mathcal{A}_0) \leftarrow 1/M$
  **for** $t = 0$ to $b - 1$ **do**
    **for** $k = 1$ to $z$ **do**
      `//Estimate model posterior with weak judge decisions`
      $\mathbb{P}(F = f_j | \mathcal{A}_t \cup \{(q, r_{(k)})\}) \leftarrow \frac{1}{Z} \mathbb{P}(F = f_j | \mathcal{A}_t) \cdot$
                            $\epsilon_{\text{loss}}^{\mathbb{1}[f_j(q) <_{(k)} \bar{f}(q)]} \epsilon_{\text{draw}}^{\mathbb{1}[f_j(q) =_{(k)} \bar{f}(q)]} \epsilon_3^{\mathbb{1}[f_j(q) >_{(k)} \bar{f}(q)]}$
    **end for**
    `//Choose the sample with minimum expected entropy`
    $q_t \leftarrow \arg\min_{q \in \mathcal{U}_t} \frac{1}{z} \sum_{k=1}^{z} \mathbb{H}(F | \mathcal{A}_t \cup \{(q, r_{(k)})\})$

    `//Get oracle decision`
    $r_t \leftarrow \text{OracleJudge}(q_t, f_j(\cdot), \bar{f}(\cdot))$
    $\mathcal{A}_{t+1} \leftarrow \mathcal{A}_t \cup \{(q_t, r_t)\}$
    $\mathcal{U}_{t+1} \leftarrow \mathcal{U}_t \setminus \{q_t\}$

    `//Update model posterior`
    $\mathbb{P}(F = f_j | \mathcal{A}_{t+1}) \leftarrow \frac{1}{Z} \mathbb{P}(F = f_j | \mathcal{A}_t) \cdot \epsilon_{\text{loss}}^{\mathbb{1}[f_j(q) < \bar{f}(q)]} \epsilon_{\text{draw}}^{\mathbb{1}[f_j(q) = \bar{f}(q)]} \epsilon_3^{\mathbb{1}[f_j(q) > \bar{f}(q)]}$
  **end for**
  **return** $\arg\max_{h \in \mathcal{M}} \text{WR}_{\mathcal{A}_b}(h, \bar{f})$

---

### 3.4 PARAMETER SELECTION

We choose the parameters $\epsilon_{\text{loss}}$ and $\epsilon_{\text{draw}}$ prior to LLM selection, therefore the oracle annotations are not available during parameter optimization. As a replacement, we use ensemble of all weak judges as a noisy oracle. More specifically, the noisy oracle behaves as follows:

$$\text{WeakJudges}(q, f(q), \bar{f}(q)) = \begin{cases} 1 & \text{if } \nu \geq 2/3 \quad // \text{ win} \\ 0.5 & \text{if } 2/3 > \nu \geq 1/3 \quad // \text{ draw} \\ 0 & \text{otherwise} \quad // \text{ loss} \end{cases}$$

where $\nu = \frac{1}{z} \sum_{k=1}^{z} \mathbb{1}[f(q) >_{(k)} \bar{f}(q)] + \frac{1}{2} \cdot \mathbb{1}[f(q) =_{(k)} \bar{f}(q)]$.

We perform a grid search over $\epsilon_{\text{loss}}$ and $\epsilon_{\text{draw}}$ using the weak judge decisions as the ground-truth annotations. We select the parameter set that maximizes the identification probability, defined as the probability of correctly recognizing the best LLM under the budget $b$.

## 4 EXPERIMENTS

In our experiments, we evaluate the effectiveness of strategies using LLM-as-a-Judge as the oracle. LLM-based evaluation serves as a reliable oracle, demonstrating strong correlation with human judgment. Moreover, LLM annotations maintain high efficiency while providing consistent and trustworthy feedback, making them a scalable and practical choice for our evaluation setting.

## 4.1 DATASET AND MODEL COLLECTIONS

We conduct experiments on several datasets including AlpacaEval (Li et al., 2023), Arena-Hard (Li et al., 2024c), and MT-Bench (Zheng et al., 2023), which contain general user dialogues; Flickr30k (Young et al., 2014) and Bingo (Cui et al., 2023), which are vision–language datasets; and MediQA (Ben Abacha et al., 2019), which focuses on medical question answering. AlpacaEval consists of 805 queries, on which we evaluate 53 LLMs. Arena-Hard contains 500 queries, with evaluations conducted on 68 LLMs. MT-Bench comprises 80 multi-turn dialogues, and we assess 6 LLMs. For Flickr30k, we use 1,000 test samples and evaluate 51 LLMs. Bingo includes 762 samples, with evaluations over 31 LLMs. Finally, MediQA contains 150 samples, on which we evaluate 9 LLMs.

Candidate models include LLMs from diverse families, including proprietary systems such as GPT-3.5 and GPT-4 (Ouyang et al., 2022b; OpenAI, 2023a), Claude 2/3 (Anthropic, 2023; 2024), and Gemini (Google, 2023), as well as open-weight architectures like LLaMA-2/3 (Touvron et al., 2023; Meta AI, 2024), Mistral and Mixtral (Jiang et al., 2023; 2024), Falcon (Almazrouei et al., 2023), Yi (Young et al., 2024), Qwen (Bai et al., 2023), Gemma (Google, 2024), InternLM (InternLM, 2023), GLM (Du et al., 2022), and DBRX (Databricks, 2024). We further consider several widely adopted instruction-tuned derivatives, including Alpaca (Taori et al., 2023), Vicuna (Chiang et al., 2023), Guanaco (Dettmers et al., 2023), Tulu-2 (Ivison et al., 2023), WizardLM (Xu et al., 2024), Zephyr (Tunstall et al., 2024), and Starling (Zhu et al., 2024). The chosen LLMs differ in both number of parameters and training methodology.

For text-only benchmarks, we employ GPT-4 (OpenAI, 2023b) as the oracle judge. For vision–language benchmarks, we rely on Prometheus-Vision (Lee et al., 2024a) as the oracle judge. For each dataset, we adopt the following baseline LLMs. AlpacaEval, Arena-Hard, and MT-Bench follow the baselines specified by their respective benchmarks: `text_davinci-003`, `gpt-4-0314`, and `gpt-3.5-turbo`, respectively. For the remaining datasets, we select as the baseline the LLM that achieves the highest performance under noisy annotations from WeakJudges. Specifically, we use `gemini-1.5-pro-preview-0514` for Flickr30k, and `gpt-4o-2024-05-13` for both Bingo and MediQA.

We choose the parameters $\epsilon_{\text{loss}}$ and $\epsilon_{\text{draw}}$ independently for each dataset, based on the procedure described in Section 3.4. Based on preliminary analysis, we set the number of weak judges $z$ to 10 in all experiments, as additional weak judges beyond this number are highly correlated with the existing ones and provide little new information.

## 4.2 BASELINES

We evaluate LLM SELECTOR against a set of baseline strategies we adapt for the active model selection task.

**Random.** At time $t$, the query $q_t$ is selected uniformly from the unannotated set $\mathcal{U}_t$: $q_t \sim \text{Uniform}(\mathcal{U}_t)$.

**Bradley-Terry.** Bradley–Terry coefficients (Bradley & Terry, 1952) are computed using annotated queries $\mathcal{A}_t$ to model LLM performances, which defines a posterior distribution over the best model. In our setting, we leverage this posterior together with entropy minimization strategy. The next query is selected by greedily minimizing the posterior entropy: $q_t = \arg\min_{q \in \mathcal{U}_t} \frac{1}{z} \sum_{k=1}^{z} \mathbb{H}(F | \mathcal{A}_t \cup \{(q, r_{(k)})\})$

**Most Draws.** For each $q \in \mathcal{U}_t$, let $d(q)$ denote the number of responses that result in a draw with the baseline response according to the ensemble WeakJudges. The next query is selected as $q_t = \arg\max_{q \in \mathcal{U}_t} d(q)$.

**Uncertainty.** We adapt the uncertainty-based sampling method of Dagan & Engelson (1995) to our setting. For each $q \in \mathcal{U}_t$, let $w(q)$, $\ell(q)$, and $d(q)$ denote the expected number of wins, losses, and draws against the baseline, as estimated by the ensemble WeakJudges. Normalizing by the total number of responses $n_q$, we obtain the empirical outcome distribution $\pi_q = (\frac{w(q)}{n_q}, \frac{\ell(q)}{n_q}, \frac{d(q)}{n_q})$. The next query is selected as the one with the highest entropy: $q_t = \arg\max_{q \in \mathcal{U}_t} \mathbb{H}(\pi_q)$.

**Confidence.** Using the same distribution $\pi_q$, the next query is selected as the one with the lowest entropy: $q_t = \arg\min_{q \in \mathcal{U}_t} \mathbb{H}(\pi_q)$.

Among the baseline strategies, only Bradley-Terry is adaptive, as its selection rule depends on the observed annotations. The remaining strategies are non-adaptive.

### 4.3 EXPERIMENTAL SETUP

In our experiments, we uniformly sample a pool of $n$ examples from the test set. We run model selection strategies on the sampled pool to select $b$ queries to annotate, and choose the LLM with the highest average utility on the annotated queries. We call this process a *realization*, and we evaluate selection strategies on multiple realizations to obtain a performance estimate.

We compare strategies by three metrics. *Identification probability* is defined as the ratio of experiments that correctly find the best model for a given budget $b$. We present results for $b = 1, \ldots, n$. *Annotation efficiency* refers to the percentage reduction in the number of labels needed to identify the best or reach within a $\delta$ vicinity of the best model across all realizations. *95th Percentile Win Rate Gap* is the 95th percentile of the win rate difference of the chosen LLMs, compared to the absolute best LLM.

### 4.4 RESULTS

#### 4.4.1 IDENTIFICATION PROBABILITY

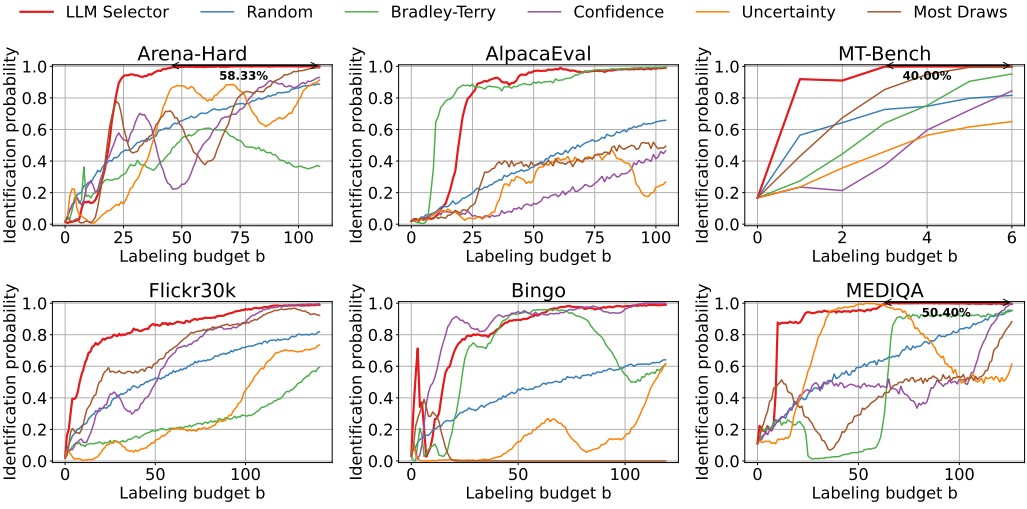

Figure 2: Best model identification probability of LLM SELECTOR and the baselines.

We present the best model identification probability of LLM SELECTOR and baseline methods in Figure 2. On Arena-Hard, MediQA, and MT-Bench, LLM SELECTOR achieves 100% identification probability with 58.33%, 50.40%, and 40.00% fewer annotated queries compared to the best competing baseline, respectively. On the remaining benchmarks, LLM SELECTOR requires a similar number of labels as the strongest baseline method. Across most values of $b$, LLM SELECTOR attains higher or comparable identification probability relative to the baselines. In contrast, baseline methods exhibit inconsistent performance: for example, Bradley–Terry performs well on AlpacaEval but is not competitive on other datasets, while Confidence performs strongly on Bingo but poorly elsewhere. By comparison, LLM SELECTOR demonstrates consistently competitive performance across all benchmarks.

#### 4.4.2 ANNOTATION EFFICIENCY FOR NEAR-BEST MODELS

Table 1 shows the annotation efficiency of LLM Selector to recover the near-best models on all datasets. We compute annotation efficiency as the relative reduction in the percentage of required annotations compared to the best competing baseline, when selecting a model within $\delta$ vicinity of

| Dataset | $\delta = 1\%$ | $\delta = 2.5\%$ | $\delta = 5\%$ |
|---|---|---|---|
| Arena-Hard | ↓ **59.62**% | ↓ **59.62**% | ↓ **58.42**% |
| AlpacaEval | ↑ 7.06% | ↓ **30.99**% | ↓ **35.85**% |
| MT-Bench | ↓ **40.00**% | ↓ **40.00**% | ↓ **42.68**% |
| Flickr30k | ↓ **3.39**% | ↓ **6.25**% | ↓ **36.47**% |
| Bingo | ↓ **7.69**% | ↓ **10.10**% | ↑ 6.00% |
| MEDIQA | ↓ **13.70**% | ↓ **6.00**% | = 0.00% |

Table 1: Annotation efficiency for near-best models across datasets: bolded numbers with ↓ indicate decreases.

the best LLM. Specifically, we measure the annotation cost saved by LLM SELECTOR to return a model within 1%, 2.5%, and 5% win rate of the best model.

We observe that LLM SELECTOR is highly annotation efficient, reaching high-performing models faster than best competing baseline. On Arena-Hard and MT-Bench, LLM SELECTOR is able to reach the top 1% and 2.5% vicinity of the best model with relatively few annotations, showing that it can reliably identify near-optimal models with limited annotation effort. On Flickr30k, Bingo, and MEDIQA LLM SELECTOR still manages to reduce the number of required annotations compared to alternative strategies, even though the improvement is smaller. LLM SELECTOR also maintains robustness under $\delta = 5\%$, using up to 58.42% fewer annotations.

### 4.4.3 ROBUSTNESS ANALYSIS

| Dataset Ident. prob. | LLM Selector (80%/90%/95%/100%) | Random (80%/90%/95%/100%) | Bradley-Terry (80%/90%/95%/100%) | Confidence (80%/90%/95%/100%) | Uncertainty (80%/90%/95%/100%) | Most Draws (80%/90%/95%/100%) |
|---|---|---|---|---|---|---|
| Arena-Hard | **11.75**/**8.13**/**0.00**/**0.00** | 13.38/13.12/11.38/8.25 | 13.00/13.00/13.00/7.87 | 13.25/13.00/14.25/9.62 | 14.12/14.00/12.25/6.87 | 12.87/13.25/12.62/7.25 |
| AlpacaEval | 4.57/**3.14**/**0.00**/**0.00** | 8.21/7.86/6.50/2.93 | 3.93/3.71/2.93/**0.00** | **3.50**/3.29/3.29/2.14 | 8.36/**3.14**/3.21/2.71 | 11.36/8.71/5.93/2.79 |
| MT-Bench | **12.50**/**12.50**/**0.00**/**0.00** | 34.17/34.17/16.67/14.17 | 33.33/33.33/16.67/**0.00** | 34.17/34.17/32.50/**0.00** | 34.17/34.17/30.83/15.83 | 52.50/52.50/17.50/**0.00** |
| Flickr30k | **6.20**/3.30/**0.00**/**0.00** | 8.40/6.10/5.30/0.00 | 9.60/7.30/6.70/0.00 | 6.60/5.00/3.90/0.00 | 11.00/6.40/5.70/0.00 | 8.40/6.00/5.40/0.00 |
| Bingo | 5.33/2.83/**0.00**/**0.00** | 8.08/6.50/6.17/3.58 | **4.00**/2.58/0.00/4.00 | 6.50/0.00/2.50/0.00 | 11.67/4.33/3.75/1.83 | 7.75/18.33/6.08/3.67 |
| MEDIQA | 2.14/**1.07**/**0.00**/**0.00** | 3.21/2.86/2.86/1.07 | 3.21/3.21/3.21/0.36 | 3.21/3.21/1.07/1.07 | 3.93/3.21/1.07/**0.00** | **1.07**/**1.07**/1.07/0.71 |

Table 2: 95th percentile win rate gap (%) at budget needed to reach identification probability 80%, 90%, 95%, and 100% on all benchmarks. Best results are in bold; second-best underlined.

To analyze the robustness, we compute the win rate gap between the selected and the true best model across all realizations. We then report the 95th percentile of these gaps, capturing the error that is larger than 95% of the observed outcomes. We perform this evaluation with varying annotation budgets, where the budgets are chosen as the amounts required for LLM SELECTOR to reach 80%, 90%, 95%, and 100% identification probability.

Table 2 shows 95th percentile win rate gap between the chosen and the best LLMs. LLM SELECTOR achieves a smaller accuracy gap when measured against the budget required to reach 80%, 90%, 95%, and 100% identification probability across nearly all datasets. The accuracy gap of LLM SELECTOR is either the best among all strategies or, the second best. Our results show that LLM SELECTOR consistently select best or near-best models with high confidence for the majority of time.

## 5 DISCUSSION

We study the novel problem of active model selection for LLMs. We adapt several baselines and introduce LLM SELECTOR, the first strategy tailored to this task. To further reduce supervision, we propose a judge-based oracle annotation scheme. Our experiments show that LLM SELECTOR lowers annotation costs while reliably identifying the best LLM across tasks and datasets. Designed for settings with scarce annotations and evolving data, LLM SELECTOR enables adaptive, cost-efficient, and robust model selection for LLM deployment.

**Ethics statement.** This work uses only publicly available datasets and models, with all sources properly cited and used according to their licenses. We introduce the first framework for active model selection of LLMs. We believe this work poses no ethical risks.

**Reproducibility statement.** We prioritize making our work easy to reproduce. All datasets we used are publicly available, and we provide the code and instructions needed to recreate every result we report. We include all code and documentation in the supplementary material so that others can reproduce our results and use our work for future comparisons.

**Large Language Model Usage.** Large Language Models helped improve the writing quality by correcting grammar mistakes, fixing typos, and enhancing text flow. The models were used only for language improvement and had no impact on the technical content, study design, or result interpretation.

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

## A  DATASET AND MODEL COLLECTIONS

For Arena-Hard, AlpacaEval, and MT-Bench, we use the available model responses along with the human judgment annotations provided by the respective benchmarks. For Flickr30k and Bingo, we conduct experiments using the data published by the VHELM (Lee et al., 2024b) benchmark. For MEDIQA, we use the data released by the MedHELM (Bedi et al., 2025) benchmark.

| Dataset | Best $\epsilon_{\text{loss}}$ across 1,000 realizations | Best $\epsilon_{\text{draw}}$ across 1,000 realizations | Dataset size | Realization pool size | LLM win rates | Number of LLMs |
|---|---|---|---|---|---|---|
| Arena-Hard | 0.20 | 0.40 | 500 | 400 | 5.20% - 84.70% | 68 |
| AlpacaEval | 0.20 | 0.40 | 805 | 700 | 15.22% - 97.64% | 53 |
| MT-Bench | 0.15 | 0.35 | 80 | 60 | 5.63% - 81.88% | 6 |
| Flickr30k | 0.40 | 0.20 | 1000 | 500 | 17.25% - 64.85% | 51 |
| Bingo | 0.60 | 0.20 | 1000 | 600 | 0.13% - 55.91% | 31 |
| MEDIQA | 0.15 | 0.35 | 150 | 140 | 33.67% - 51.00% | 9 |

Table 3: Summary of the six datasets used in our experiments.

Table 3 provides an overview of the six datasets used in our experiments, including the best $\epsilon_{\text{loss}}$ and $\epsilon_{\text{draw}}$ across 1,000 realizations, dataset sizes, realization pool sizes, ranges of model win rates, and the number of pretrained models. The datasets vary in size, number of available models, and ranges of model win rates, allowing us to evaluate our methods under diverse experimental scenarios.

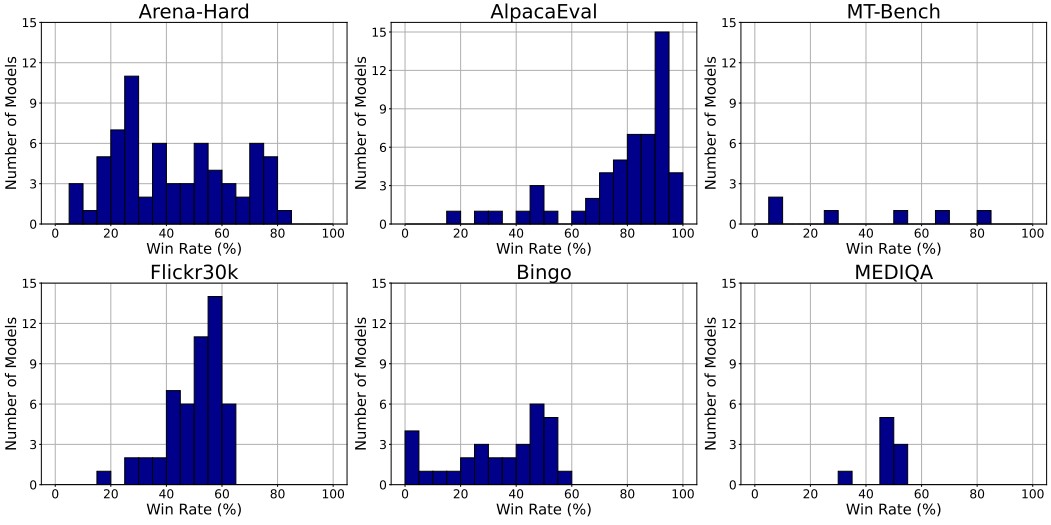

Figure 3: Candidate LLM win rate histograms.

The performance of the candidate LLMs is plotted in Figure 3. The plots show the histogram of models which are in the different win rate ranges for each dataset. The histograms show that experiments include a diverse range of win rates against the baseline. This indicates that our experiments cover different scenarios and capture the variability present in real-world applications.

## B  ANALYSIS OF ALTERNATIVE WEAK JUDGES

Figure 4 presents the performance of LLM SELECTOR when using two additional weak judges based on sentence embedding models and reward models. As shown, $k$-gram weak judges outperform both alternatives, highlighting the effectiveness of our $k$-gram-based approach.

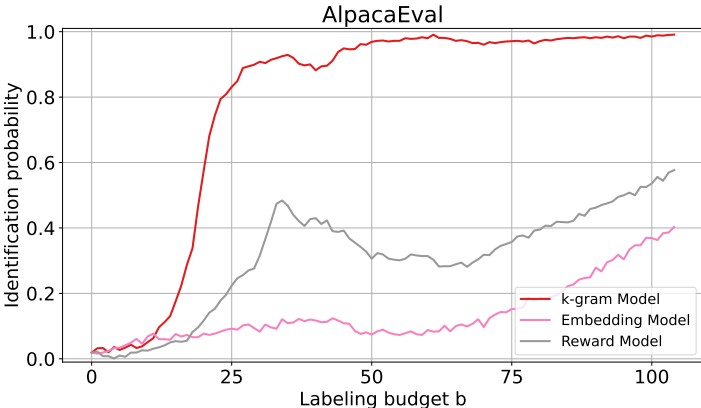

Figure 4: Comparison of the performance of LLM SELECTOR with k-gram, embedding, and reward model based weak judges.

Specifically, the embedding-based weak judge computes sentence embeddings for all responses to a given query using the `all-mpnet-base-v2` model from the Sentence Transformers library (Reimers & Gurevych, 2019). Then, the mean embedding across responses is calculated, and for any pair of responses, the weak judge prefers the response whose embedding is closer to this mean. On the other hand, the reward-model-based weak judge directly uses the pairwise comparison outputs of a reward model to make decisions. In particular, we employ `Skywork-Reward-V2-Llama-3.1-8B` (Liu et al., 2025) as the weak judge.

## C  DERIVATION OF THE POSTERIOR UPDATE

We provide here a derivation of the posterior update shown in Equation 4, starting from the likelihood model defined in Equation 2.

The probability in Equation 4 represents the posterior probability of each candidate LLM being the best model, conditioned on the oracle judge annotations on $A_t$ and the weak judge annotation on $q$. We compute this posterior via Bayes' rule. Specifically, for each model $f_j$, the posterior is

$$\mathbb{P}(F = f_j \mid A_t \cup \{(q, R = r_{(k)})\}) \propto \mathbb{P}(A_t \cup \{(q, R = r_{(k)})\} \mid F = f_j) \cdot \mathbb{P}(F = f_j).$$

Assuming conditional independence of the oracle annotations in $A_t$ and the weak annotation on $q$ given the model $f_j$, the posterior factorizes as

$$\mathbb{P}(F = f_j \mid A_t \cup \{(q, R = r_{(k)})\}) \propto \mathbb{P}(\{(q, R = r_{(k)})\} \mid F = f_j) \cdot \mathbb{P}(F = f_j \mid A_t).$$

Using the likelihood defined in Equation 2, the posterior can be written as

$$\mathbb{P}(F = f_j | A_t \cup \{(q, r_{(k)})\}) \propto \epsilon_{\text{loss}}^{\mathbb{1}[f_j(q) <_{(k)} \bar{f}(q)]} \cdot \epsilon_{\text{draw}}^{\mathbb{1}[f_j(q) =_{(k)} \bar{f}(q)]}$$
$$\cdot (1 - \epsilon_{\text{loss}} - \epsilon_{\text{draw}})^{\mathbb{1}[f_j(q) >_{(k)} \bar{f}(q)]} \mathbb{P}(F = f_j | A_t)$$

This yields the expression shown in Equation 4.

