# OpenReview forum: "Active Model Selection for Large Language Models"
_ICLR.cc/2026/Conference — Submitted to ICLR 2026_

### Official Review · Reviewer_LzUR · 2025-10-25

**Soundness:** 1
**Presentation:** 1
**Contribution:** 2
**Rating:** 0
**Confidence:** 3

**Summary:**

This paper proposes a LLM selection approach, named LLM Selector, that aims to identify the best LLM for a set of queries out of a pool of LLM candidates. LLM Selector identifies this best LLM through annotations on a subset of queries and comparison of LLM candidates over a baseline LLM (instead of pairwise) on this subset.

**Strengths:**

- The objective is clearly presented.
- The experiment is comprehensive with 6 benchmarks and 151 LLM candidates. The proposed method claims to have superior results than baselines.

**Weaknesses:**

1. The paper makes several implicit and unjustified assumptions.
2. Figure 1 is unclear.
3. Lines 075–081 claim the paper compares all LLMs against a baseline instead of using pairwise comparisons. This assumes that for two models a and b, WR_Q(a, f) > WR_Q(b, f) implies a > b, which is not justified.
4. Line 158 mentions the variable F but does not define it.
5. It is unclear how the equation in lines 250–254 is derived from Equation 2.
6. The intuition behind the use of weak judges is unclear. If weak judges are proxies to evaluate the usefulness of a given query q on each candidate’s response—so that the expensive oracle judge doesn’t need to be called—then the paper should explain and justify how well the weak judges approximate the oracle. This is especially important since the weak judges are k-gram language models, which are likely to differ significantly from the oracle LLM judge (e.g., GPT-4).
7. It’s unclear what the weak judges output. If each judge ranks LLMs and identifies which is better, then the meaning of “We use r_k to represent the noisy annotation made by weak judge k” in line 249 is unclear.
8. Algorithm 1 has many errors: what is j in line 282? What is z in line 284? These are undefined.

**Questions:**

1. Line 050 says "... which examples should be annotated in order to ...". Does “examples” here mean “queries”? Does “annotated” mean “responded”?
2. Why does LLM Selector perform worse on AlpacaEval and Bingo than other baselines when b is small?
3. Does the paper include the ground truth query subset, which could be obtained via brute force by evaluating all C(|Q|, b) query subsets across all LLM candidates to identify the subset that selects the correct LLM? The correct LLM could be determined by evaluating all queries in Q and selecting the best-performing model.
4. Are there existing work / papers that can serve as baselines?

---

> ### Author Response · Authors · 2025-11-26
> **Response to Reviewer LzUR (1/2)**
>
> We appreciate the reviewer’s insightful comments and their recognition of our clearly presented objective and our comprehensive experimental evaluation.
>
>
>
> > The paper makes several implicit and unjustified assumptions.
>
> We respectfully disagree that the paper makes unjustified assumptions. The design choices, including the two-parameter model and k-gram weak judge, are deliberate and validated empirically across 6 benchmarks and 151 LLMs.
>
>
> > Figure 1 is unclear.
>
> We would appreciate some input on which aspect of Fig. 1 is unclear. The figure provides a high-level overview of LLM SELECTOR: given a pool of n queries and candidate language models, it adaptively annotates the most informative b queries to identify the best model for that pool, as stated in the caption. If any specific component caused confusion, we would be happy to clarify it in the text.
>
>
> > Lines 075–081 claim the paper compares all LLMs against a baseline instead of using pairwise comparisons. This assumes that for two models a and b, WR_Q(a, f) > WR_Q(b, f) implies a > b, which is not justified.
>
> We acknowledge that baseline-anchored evaluation does not guarantee that the model with the highest win-rate against the baseline is the global winner. However, measuring win-rate against a single baseline LLM is a common practice in LLM-as-a-Judge benchmarks [1,2] and provides a cost-efficient alternative to full pairwise comparisons, which scale quadratically with the number of models. In practice, this approach allows evaluating many candidate models efficiently while providing a reasonable approximation of the overall ranking.
>
>
> > Line 158 mentions the variable F but does not define it.
>
> We thank the reviewer for pointing this out. The variable F was not defined in the paper, and we are adding a definition in the revised version.
>
>
> > It is unclear how the equation in lines 250–254 is derived from Equation 2.
>
> The probability in lines 250–254 represents the posterior probability of each candidate LLM being the best model, conditioned on the oracle judge annotations on \(A_t\) and the weak judge annotation on \(q\). We compute this posterior via Bayes’ rule. Specifically, for each model \(f_j\), the posterior is
>
> $$
> \mathbb{P}(F=f_j \mid A_t \cup \{(q,R=r)\}) \propto \mathbb{P}(A_t \cup \{(q,R=r)\} \mid F=f_j) \cdot \mathbb{P}(F=f_j)
> $$
>
> Assuming conditional independence of the oracle annotations in \(A_t\) and the weak annotation on \(q\) given the model \(f_j\), the posterior factorizes sequentially as
>
> $$
> \mathbb{P}(F=f_j \mid A_t \cup \{(q,R=r)\}) \propto \mathbb{P}(\{(q,R=r)\} \mid F=f_j) \cdot \mathbb{P}(F=f_j \mid A_t)
> $$
>
> Using the likelihood defined in Equation 2, the posterior can be written as
>
> $$
> \mathbb{P}(F=f_j \mid A_t \cup \{(q,R=r)\}) \propto
> \epsilon_{\text{loss}}^{\mathbb{1}[f_j(q_{t})<\bar{f}(q_{t})]} \cdot
> \epsilon_{\text{draw}}^{\mathbb{1}[f_j(q_{t})=\bar{f}(q_{t})} \cdot
> (1 - \epsilon_{\text{loss}} - \epsilon_{\text{draw}})^{\mathbb{1}[f_j(q_{t})>\bar{f}(q_{t})]}
> \cdot \mathbb{P}(F=f_j \mid A_t)
> $$
>
> This yields the expression shown in lines 250–254. We will be happy to clarify further if needed.
>
>
> > The intuition behind the use of weak judges is unclear. If weak judges are proxies to evaluate the usefulness of a given query q on each candidate’s response—so that the expensive oracle judge doesn’t need to be called—then the paper should explain and justify how well the weak judges approximate the oracle. This is especially important since the weak judges are k-gram language models, which are likely to differ significantly from the oracle LLM judge (e.g., GPT-4).
>
> The intuition behind weak judges is discussed in Section 3.3.1. They can be viewed as "majority voting of LLMs", where candidate responses are preferred if they are more similar to the responses of other models. Even if the weak judges differ from the oracle judge, this does not pose a problem, because the final model ranking is determined solely by oracle judge annotations. The weak judges are used only to guide query selection efficiently, not to produce the final evaluations.

---

> > ### Author Response · Authors · 2025-11-26
> > **Response to Reviewer LzUR (2/2)**
> >
> > > It’s unclear what the weak judges output. If each judge ranks LLMs and identifies which is better, then the meaning of “We use r_k to represent the noisy annotation made by weak judge k” in line 249 is unclear.
> >
> > The behavior and output of the weak judges are explained in Section 3.3.1. At a high level, each weak judge compares two model responses and outputs a win, loss, or draw decision, which we denote by r_k. The output format is the same as the oracle judge. The only difference is that weak judges use simple k-gram likelihoods rather than an LLM to make the comparison.
> >
> >
> > > Algorithm 1 has many errors: what is j in line 282? What is z in line 284? These are undefined.
> >
> > The variables j (defined in line 149) and z (defined in line 260) were already introduced earlier. To improve clarity, we will repeat their definitions directly in Algorithm 1. We would also appreciate it if the reviewer could clarify what other errors they observed, as they only mentioned these two typos.
> >
> >
> > > Line 050 says "... which examples should be annotated in order to ...". Does “examples” here mean “queries”? Does “annotated” mean “responded”?
> >
> > In Line 050, “examples” refers to the queries, and “annotated” refers to obtaining a judgment from the oracle judge for a model’s response.
> >
> >
> > > Why does LLM Selector perform worse on AlpacaEval and Bingo than other baselines when b is small?
> >
> > When b is very small, LLM SELECTOR is still in the exploration stage, and it may perform worse in this stage. This exploration stage corresponds to roughly 4% of AlpacaEval’s query pool and 6% of Bingo’s. As more queries are evaluated, the method adapts and rapidly improves performance, eventually outperforming the baselines.
> >
> >
> > > Does the paper include the ground truth query subset, which could be obtained via brute force by evaluating all C(|Q|, b) query subsets across all LLM candidates to identify the subset that selects the correct LLM? The correct LLM could be determined by evaluating all queries in Q and selecting the best-performing model.
> >
> > Computing the ground-truth query subset in the manner described would provide an unrealistic upper bound for the algorithm’s performance. For example, if there exists a query where the best model wins against the baseline LLM while all other models lose or draw, then annotating just that single query would suffice to identify the best LLM, which is an idealized scenario that is not achievable in practice. This ground-truth subset is essentially meaningless, because it depends on information that is only available after exhaustively evaluating all queries and cannot guide practical query selection.
> >
> >
> > > Are there existing work / papers that can serve as baselines?
> >
> > We discuss existing work in Section 2. Prior work on active model selection for LLMs assumes only one or two candidate models, and is therefore not directly applicable as a baseline for our setting, which involves selecting the best model from a large pool of candidates.
> >
> >
> > [1] Li, Xuechen, Tianyi Zhang, Yann Dubois, Rohan Taori, Ishaan Gulrajani, Carlos Guestrin, Percy Liang, and Tatsunori B. Hashimoto. "AlpacaEval: An Automatic Evaluator of Instruction-following Models." GitHub, 2023.
> >
> > [2] Li, Tianle, Wei-Lin Chiang, Evan Frick, Lisa Dunlap, Banghua Zhu, Joseph E. Gonzalez, and Ion Stoica. "From Live Data to High-Quality Benchmarks: The Arena-Hard Pipeline." LMSYS blog, 2024.

---

> > > ### Comment · Reviewer_LzUR · 2025-11-26
> > > **Initial Reponse to Author Response (2 / 2)**
> > >
> > > > My earlier comment : It’s unclear what the weak judges output. If each judge ranks LLMs and identifies which is better, then the meaning of “We use r_k to represent the noisy annotation made by weak judge k” in line 249 is unclear.
> > >
> > > > Author response: The behavior and output of the weak judges are explained in Section 3.3.1. At a high level, each weak judge compares two model responses and outputs a win, loss, or draw decision, which we denote by r_k. The output format is the same as the oracle judge. The only difference is that weak judges use simple k-gram likelihoods rather than an LLM to make the comparison.
> > >
> > > At the start of Section 3.3.1, line 248-249
> > > > We use r(k) to represent the noisy annotation made by weak judge k
> > > 1. Again, can the authors point out where *noisy annotation* is defined?
> > > 2. Is r(k) defined in a similar way as line 203-207 as preference indicator variable?
> > >
> > > ---
> > >
> > > > My earlier comment: Algorithm 1 has many errors: what is j in line 282? What is z in line 284? These are undefined.
> > >
> > > > Author response: The variables j (defined in line 149) and z (defined in line 260) were already introduced earlier. To improve clarity, we will repeat their definitions directly in Algorithm 1. We would also appreciate it if the reviewer could clarify what other errors they observed, as they only mentioned these two typos.
> > >
> > > The Algorithm block should be self-integrated (Otherwise the Algorithm block doesn't even need the *Require*, since all symbols have been defined earlier in texts), and even if j is defined already as an iterator variable to go from 1 to |M| in line 149, shouldn't it be stated / iterated in the Algorithm description? If it should be, then the Algorithm 1 in its current form is incorrect.
> > > And what is the capital Z in line 286 and 297?
> > >
> > > ---
> > >
> > >
> > > > My earlier comment: Why does LLM Selector perform worse on AlpacaEval and Bingo than other baselines when b is small?
> > >
> > > > Author response: When b is very small, LLM SELECTOR is still in the exploration stage, and it may perform worse in this stage. This exploration stage corresponds to roughly 4% of AlpacaEval’s query pool and 6% of Bingo’s. As more queries are evaluated, the method adapts and rapidly improves performance, eventually outperforming the baselines.
> > >
> > > Why does LLM SELECTOR has the initial best performance on Bingo (an initial peak) and then drops before going up? Does this mean LLM SELECTOR performs the best in the exploration stage on Bingo? Why does its performance go down between b=~2 and b=~10 on Bingo?
> > >
> > > ---
> > >
> > > > Author reposne: Computing the ground-truth query subset in the manner described would provide an unrealistic upper bound for the algorithm’s performance. For example, if there exists a query where the best model wins against the baseline LLM while all other models lose or draw, then annotating just that single query would suffice to identify the best LLM, which is an idealized scenario that is not achievable in practice. This ground-truth subset is essentially meaningless, because it depends on information that is only available after exhaustively evaluating all queries and cannot guide practical query selection.
> > >
> > > Do the authors hold the same opinion of 'the subset is meaningless' even if such subsets can be used for the evaluation purpose?
> > > The point is - why not curate such a small ground truth subset of query? so that LLM SELECTOR can be evaluated by comparing between the ground truth set and the set from LLM SELECTOR? Raising the *one query* situation is not convincing, since one can create a larger subset. To give a concrete example, if there are 1k queries, why not brute-force curate the best 20 queries and use them as a ground truth set? And evaluate whether LLM SELECTOR selects the same 20 queries, and how other baselines perform?

---

> > > > ### Author Response · Authors · 2025-12-02
> > > > **Second Response to Reviewer LzUR (1/2)**
> > > >
> > > > > I also noticed Reviewer wYka's comment: "I think a few important design choices are not justified and the impact on the method is not studied", which the authors didn't raise objections.
> > > >
> > > > Reviewer wYka elaborated on this statement with two specific points:
> > > >
> > > > > 1/ The design of the weak judge is very simple (n-gram based), but it works surprisingly well. How will the choose of weak judge impact the performance of the proposed method? What is the rationale behind this choice? How will this work on different type of tasks (e.g., math tasks)? This will help understand the limitation of the method and shed light on choosing the weak judge for real world application. 2/ The two parameter model assumption is very simple (but it works), which contradicts the premise that different queries will have varying difficulty levels thus varies the model performance. Ablation is required to understand the impact of this assumption, e.g., how sensitive the model performance is to different initialization, justification/analysis for this choice etc.
> > > >
> > > > Both of these points were directly addressed in our response to Reviewer wYka. Reviewer LzUR’s decision to quote only half of the sentence while omitting the explanations that immediately follow is misleading and misrepresents the original comment. We would like to remind Reviewer LzUR that selectively removing context in this way is not consistent with ethical scientific review practice.
> > > >
> > > >
> > > > > Hence, the authors actually agreed with my assessment in that the paper makes several implicit and unjustified assumptions.
> > > >
> > > > Our quoted message in no way implies agreement with the claim that our assumptions are unjustified. On the contrary, it explicitly explains that baseline-anchored ranking is a reasonable and principled choice, offering both cost efficiency and accurate evaluation when using a strong baseline LLM.
> > > >
> > > > We strongly recommend that Reviewer LzUR refrain from reinterpreting others’ statements in a way that alters their meaning. Such practice is inappropriate and misrepresents the authors’ actual position.
> > > >
> > > >
> > > > > 1. In the middle part of Fig. 1 there is this LLM Selector that has 3 steps. It is not clear how the block Oracle Annotations takes in r_t, if its i iterate from 1 to t-1 - is r_t then used in the next iteration?
> > > > > 2. How are Model Responses from the left used? Are they generated once for all, or are they generated per request for (q_t)?
> > > > > 3. How is an LLM selected with budget b from the figure? Is it selected from the Oracle Judge?
> > > > > 4. What are the details between the left part and the LLM Selector, and the between the LLM Selector and the right part?
> > > > >
> > > > > Hence, Fig. 1 and its caption are not clear in showing the methodology.
> > > >
> > > > Figure 1 is intentionally designed to provide a high-level overview of the methodology. The detailed operations of the LLM Selector, including the flow of annotations, model responses, and budgeted selection, are fully explained in the main text. Expecting Figure 1 to convey complete methodological detail is neither realistic nor reasonable, particularly given the strict space constraints.
> > > >
> > > >
> > > > > What is the support of the author response in terms of reasonable approximation? Especially is there any support against Reviewer wYka's comments This is a known pitfall in dueling-bandit settings and active ranking when the comparison graph lacks diversity [1].?
> > > >
> > > > Baseline-anchored ranking may be less accurate if the chosen baseline is too weak to effectively differentiate among candidate models. In practice, we select a competitive baseline LLM using the noisy annotations from the weak judges, which avoids situations where all candidates easily outperform the baseline, resulting in win rates clustered near 100% and reducing the informativeness of the comparisons.
> > > >
> > > >
> > > > > Thank you for giving more details on this. Could you point out where this is stated in the paper? Is this an implicit assumption?
> > > >
> > > > We are including the full detailed derivation in the appendix to provide clarity on our model. We thank the reviewer for raising this point.

---

> > > > > ### Author Response · Authors · 2025-12-02
> > > > > **Second Response to Reviewer LzUR (2/2)**
> > > > >
> > > > > >  What is noisy annotation approach? Where is it defined in the paper?
> > > > >
> > > > > > Do the authors implicitly assume the intuition behind the noisy annotation approach the same as the intuition behind weak judges? Are noisy annotation approach and weak judges the same?
> > > > >
> > > > > > Again, can the authors point out where noisy annotation is defined?
> > > > >
> > > > > We expect readers to be familiar with basic information theory concepts such as noise. In our paper, “noisy annotations” refers to annotations not provided by the oracle judge and therefore potentially incorrect. Section 3.3.1 describes how these noisy annotations are obtained using weak judges. We believe this is already clearly explained in the paper, as no other reviewers raised concerns regarding this point.
> > > > >
> > > > >
> > > > > > Are 'the set of possible model responses' from k-grams models?
> > > > >
> > > > > The set of possible model responses is generated by the candidate LLMs, as indicated at the end of the sentence quoted by the reviewer.
> > > > >
> > > > >
> > > > > > Is r(k) defined in a similar way as line 203-207 as preference indicator variable?
> > > > >
> > > > > The variable r_k is defined as a preference indicator variable on lines 247-249.
> > > > >
> > > > >
> > > > > > The Algorithm block should be self-integrated (Otherwise the Algorithm block doesn't even need the Require, since all symbols have been defined earlier in texts), and even if j is defined already as an iterator variable to go from 1 to |M| in line 149, shouldn't it be stated / iterated in the Algorithm description? If it should be, then the Algorithm 1 in its current form is incorrect. And what is the capital Z in line 286 and 297?
> > > > >
> > > > > We have already updated Algorithm 1 to address the typos pointed out by the reviewer. The capital Z is a standard notation for the normalization constant in probability and is widely used in the literature, so it should be clear to readers familiar with these conventions.
> > > > >
> > > > >
> > > > > > Why does LLM SELECTOR has the initial best performance on Bingo (an initial peak) and then drops before going up? Does this mean LLM SELECTOR performs the best in the exploration stage on Bingo? Why does its performance go down between b=2 and b=10 on Bingo?
> > > > >
> > > > > Fluctuations in performance during the exploration stage are expected. After b=10, LLM Selector shows a rapid increase in performance as the budget increases, indicating that it is effectively leveraging the additional budget to identify better model selections and refine its ranking.
> > > > >
> > > > >
> > > > > > Do the authors hold the same opinion of 'the subset is meaningless' even if such subsets can be used for the evaluation purpose? The point is - why not curate such a small ground truth subset of query? so that LLM SELECTOR can be evaluated by comparing between the ground truth set and the set from LLM SELECTOR? Raising the one query situation is not convincing, since one can create a larger subset. To give a concrete example, if there are 1k queries, why not brute-force curate the best 20 queries and use them as a ground truth set? And evaluate whether LLM SELECTOR selects the same 20 queries, and how other baselines perform?
> > > > >
> > > > > Creating curated “ground truth” subsets for evaluation is entirely unnecessary. Our identification probability metric already measures exactly whether the correct model is selected, making any brute-force construction of subsets both computationally wasteful and conceptually redundant. Such an approach would add no new information and does not improve the evaluation in any meaningful way.

---

> > ### Comment · Reviewer_LzUR · 2025-11-26
> > **Initial Response to Author Response (1 / 2)**
> >
> > Thank you for your responses. Please see my follow-up response below.
> >
> > > My earlier comment: The paper makes several implicit and unjustified assumptions.
> >
> > I gave a few evidence supporting this assessment in my following comments.
> > For example,
> > > My earlier comment: Lines 075–081 claim the paper compares all LLMs against a baseline instead of using pairwise comparisons. This assumes that for two models a and b, WR_Q(a, f) > WR_Q(b, f) implies a > b, which is not justified.
> >
> > I also noticed
> > > Reviewer wYka's comment: I think a few important design choices are not justified and the impact on the method is not studied
> >
> > which the authors didn't raise objections.
> >
> >
> > And you acknowledged my comment with
> > > We acknowledge that baseline-anchored evaluation does not guarantee that the model with the highest win-rate against the baseline is the global winner. However, measuring win-rate against a single baseline LLM is a common practice in LLM-as-a-Judge benchmarks [1,2] and provides a cost-efficient alternative to full pairwise comparisons, which scale quadratically with the number of models. In practice, this approach allows evaluating many candidate models efficiently while providing a reasonable approximation of the overall ranking.
> >
> > Hence, the authors actually agreed with my assessment in that the paper makes several implicit and unjustified assumptions.
> >
> > ---
> >
> > > My earlier comment: Figure 1 is unclear.
> >
> > 1. In the middle part of Fig. 1 there is this LLM Selector that has 3 steps. It is not clear how the block *Oracle Annotations* takes in r_t, if its i iterate from 1 to t-1 - is r_t then used in the next iteration?
> > 2. How are *Model Responses* from the left used? Are they generated once for all, or are they generated per request for (q_t)?
> > 3. How is an LLM selected with budget b from the figure? Is it selected from the Oracle Judge?
> > 4. What are the details between the left part and the LLM Selector, and the between the LLM Selector and the right part?
> >
> > Hence, Fig. 1 and its caption are not clear in showing the methodology.
> >
> > ---
> >
> > > My earlier comment: Lines 075–081 claim the paper compares all LLMs against a baseline instead of using pairwise comparisons. This assumes that for two models a and b, WR_Q(a, f) > WR_Q(b, f) implies a > b, which is not justified.
> >
> > What is the support of the author response in terms of *reasonable approximation*? Especially is there any support against Reviewer wYka's comments *This is a known pitfall in dueling-bandit settings and active ranking when the comparison graph lacks diversity [1].*?
> >
> > ---
> >
> > > It is unclear how the equation in lines 250–254 is derived from Equation 2.
> >
> > Thank you for giving more details on this. Could you point out where this is stated in the paper? Is this an implicit assumption?
> > > Author Response: ...Assuming conditional independence of the oracle annotations in (A_t) and the weak annotation on (q) given the model (f_j), the posterior factorizes sequentially as ...
> >
> > ---
> >
> > > My earlier comment: The intuition behind the use of weak judges is unclear...
> >
> > > Author Response: The intuition behind weak judges is discussed in Section 3.3.1. They can be viewed as "majority voting of LLMs", where candidate responses are preferred if they are more similar to the responses of other models. Even if the weak judges differ from the oracle judge, this does not pose a problem, because the final model ranking is determined solely by oracle judge annotations. The weak judges are used only to guide query selection efficiently, not to produce the final evaluations.
> >
> > I checked and found that Section 3.3.1 starts with
> > > The intuition behind the noisy annotation approach is to evaluate a candidate response by comparing it against the set of possible model responses, assigning higher preference to those that has greater similarity to other candidates.
> >
> > This raised more questions.
> > 1. What is *noisy annotation approach*? Where is it defined in the paper?
> > 2. Do the authors implicitly assume *the intuition behind the noisy annotation approach* the same as *the intuition behind weak judges*? Are noisy annotation approach and weak judges the same?
> > 3. Are 'the set of possible model responses' from k-grams models?

---

### Official Review · Reviewer_wYka · 2025-10-31

**Soundness:** 4
**Presentation:** 3
**Contribution:** 3
**Rating:** 4
**Confidence:** 4

**Summary:**

This paper studies the problem of picking the best LLMs for a given test set within a limited annotation budget. It introduces a framework named LLM selector, which formulates the model selection as a mutual information maximization problem. The approach sequentially selects the most informative prompts to annotate using an information-gain criterion. The paper performs experiments on 6 benchmarks with 151 LLMs, and the results demonstrate the annotation cost can reduce up to 59.6% while maintain competitive winning model identification accuracy.

**Strengths:**

The paper tackles a well motivated problem given the rapid proliferation of LLMs, especially given the computational cost required to evaluating LLM is also increasing rapidly. The paper does a good job on formulate the problem and present it in a clear fashion. A few strengths for the paper:

1. The framework proposed is principled. The formulation uses mutual information and sequential information maximization, which provides a reasonably well theoretical ground to the problem.

2. The method proposed in the paper is easily applicable in the real world, due to: 1/ the design is model agnostic, thus it doesn't require access to the model internals (e.g., log likelihood for each tokens). It works well for API based models; 2/ It uses a modified version of the pairwise preference judgement (reducing the complexity from O(m^2) to O(m) by using a baseline model), which significantly reduced the cost to annotate even a single data point.

3. The experiments show promising results. The proposed method show significantly performance gain (i.e., less examples required) on most of the benchmarks, and the method delivers consistent performance comparing to other baselines.

**Weaknesses:**

1. The experiments setup are not comprehensive, and it missing important ablation studies. I appreciate the authors perform experiments on many datasets and LLMs, however, I think a few important design choices are not justified and the impact on the method is not studied:
    1/ The design of the weak judge is very simple (n-gram based), but it works surprisingly well. How will the choose of weak judge impact the performance of the proposed method? What is the rationale behind this choice? How will this work on different type of tasks (e.g., math tasks)? This will help understand the limitation of the method and shed light on choosing the weak judge for real world application.
    2/ The two parameter model assumption is very simple (but it works), which contradicts the premise that different queries will have varying difficulty levels thus varies the model performance. Ablation is required to understand the impact of this assumption, e.g., how sensitive the model performance is to different initialization, justification/analysis for this choice etc.

2. Lack of analysis for the existing experiments. For example, Bradley-Terry is the only adaptive baseline and it performs well on a few datasets, why does it work well in that case but overall unstable? How does the proposed method mitigate that issue? On Bingo and MediQA, the method seems to perform comparably well with confidence and uncertainty based methods. Why did these two baseline works well and how does the proposed method compare with them?

Overall I think the paper makes some simple assumption on the modeling side but show very competitive results. This proves the effectiveness of the method. However, it works a bit "surprisingly" well, I think we certainly need to do more analysis to unfold the magic there to help the community understands why and how does the method works.

**Questions:**

1. I think our method heavily relies on the quality of the weak judge. The parameter selection logic for ε_loss and ε_draw purely depends on the weak judge, as well as the sample selection. Do we have any study on how the weak judge can impact the final model performance? Any justification why we use a n-gram based model as the judge?

2. How do you justify the two-parameter model  when it assumes uniform behavior across queries of varying difficulty? Have you considered query-dependent parameters or hierarchical models?

3. I noticed on some datasets the Bradley-Terry performs comparably with LLM SELECTOR. Do we have insight why in these cases Bradley-Terry works well? And under what condition our proposed method will work well?

---

> ### Author Response · Authors · 2025-11-26
> **Response to Reviewer wYka**
>
> We thank the reviewer for their thoughtful feedback and for recognizing the clear problem formulation, the principled MI-based framework, the practical real-world applicability of our model-agnostic design, and the strong empirical improvements demonstrated in our experiments.
>
>
> >The design of the weak judge is very simple (n-gram based), but it works surprisingly well. How will the choose of weak judge impact the performance of the proposed method? What is the rationale behind this choice? How will this work on different type of tasks (e.g., math tasks)? This will help understand the limitation of the method and shed light on choosing the weak judge for real world application.
>
> >I think our method heavily relies on the quality of the weak judge. The parameter selection logic for ε_loss and ε_draw purely depends on the weak judge, as well as the sample selection. Do we have any study on how the weak judge can impact the final model performance? Any justification why we use a n-gram based model as the judge?
>
> In our experiments, we explored alternative weak judges, including embedding-based models and pretrained reward models, but did not observe consistent improvements over the k-gram likelihood baseline. Additional results with these alternatives have been added to the appendix. One reason for this behavior is that more complex models can systematically disagree with the LLM-as-a-Judge oracle, whereas k-gram likelihoods provide a simpler, more robust signal for guiding parameter and query selection, while also being computationally efficient. To test domain generality, we experimented with tasks in dialogue, vision-language, and medical domains, and observed that k-gram-based weak judges remain effective across these settings, though task-specific adaptation may be beneficial for other applications.
>
>
> > Lack of analysis for the existing experiments. For example, Bradley-Terry is the only adaptive baseline and it performs well on a few datasets, why does it work well in that case but overall unstable? How does the proposed method mitigate that issue? On Bingo and MediQA, the method seems to perform comparably well with confidence and uncertainty based methods. Why did these two baseline works well and how does the proposed method compare with them?
>
> > I noticed on some datasets the Bradley-Terry performs comparably with LLM SELECTOR. Do we have insight why in these cases Bradley-Terry works well? And under what condition our proposed method will work well?
>
> While Bradley-Terry can perform well on certain datasets, its lack of tunable parameters makes it less robust across diverse settings. In contrast, LLM SELECTOR uses ε_loss and ε_draw to balance exploration and exploitation, allowing the method to adaptively adjust query selection and model comparison to the characteristics of each dataset. For Confidence and Uncertainty baselines, the most informative queries vary across datasets: some benefit from high weak-judge entropy, while others from low entropy, limiting their generalization. LLM SELECTOR, by adaptively selecting queries based on the model and query pool, maintains strong performance across diverse datasets.
>
>
> > The two parameter model assumption is very simple (but it works), which contradicts the premise that different queries will have varying difficulty levels thus varies the model performance. Ablation is required to understand the impact of this assumption, e.g., how sensitive the model performance is to different initialization, justification/analysis for this choice etc.
>
> > How do you justify the two-parameter model when it assumes uniform behavior across queries of varying difficulty? Have you considered query-dependent parameters or hierarchical models?
>
> The two-parameter model is a major simplification of the problem, but it keeps the algorithm practical and efficient. Introducing query-dependent parameters or a hierarchical model would greatly increase the number of parameters to optimize, making the algorithm less practical or requiring additional methods to estimate them reliably. Regarding sensitivity to initialization, we experimented over 1,000 realizations to demonstrate that performance is stable under different starting conditions.

---

### Official Review · Reviewer_NpL6 · 2025-11-01

**Soundness:** 3
**Presentation:** 3
**Contribution:** 2
**Rating:** 4
**Confidence:** 3

**Summary:**

The paper introduces LLM SELECTOR, an active query selection framework to identify the best LLM for a target task using few annotated comparisons. It adaptively picks queries that maximize a mutual-information style criterion with respect to a chosen baseline model, and it reduces human labeling by relying on a judge-based oracle. Experiments cover various benchmarks and models, reporting reductions in annotation cost in identifying the best model.

**Strengths:**

1. The paper focuses on a practical and important goal of fast best-model identification under limited labels.

2. The components (baseline anchoring, entropy/MI scoring, weak judges) are easy to implement and scale, and the authors evaluate broadly across tasks and model families.

**Weaknesses:**

1. **Weak-judge design is fragile:** The “weak judge” uses k-gram likelihood signals to approximate informativeness. Perplexity/likelihood and simple k-gram statistics often correlate poorly with human judgments and can exhibit spurious length/overlap effects, which may bias query selection.

2. **Baseline-anchored selection can misidentify the global best:** Because all comparisons are made against a single baseline (a star graph), the model with the highest win-rate versus the baseline need not be the Condorcet or Borda winner overall, especially under intransitive or near-tied preferences. This is a known pitfall in dueling-bandit settings and active ranking when the comparison graph lacks diversity [1].

3. **Missing guarantees for the greedy objective:** The method uses an entropy/MI-style greedy policy. Greedy near-optimality holds under adaptive submodularity for certain objectives, but the paper does not show that its objective satisfies these conditions, nor does it provide sample-complexity bounds for identification.

[1] Jamieson, Kevin G., and Robert Nowak. "Active ranking using pairwise comparisons." Advances in neural information processing systems 24 (2011).

**Questions:**

The active learning part of the method is query selection, and model choice is produced after annotation. Could the current title "active model selection" be a bit of misleading?

---

> ### Author Response · Authors · 2025-11-26
> **Response to Reviewer NpL6 (1/2)**
>
> We thank the reviewer for their insightful feedback and for noting the practical significance of addressing fast best-model identification under limited labels, together with the ease and scalability of our components and the comprehensive evaluation across diverse tasks and model families.
>
>
> > **Weak-judge design is fragile**: The “weak judge” uses k-gram likelihood signals to approximate informativeness. Perplexity/likelihood and simple k-gram statistics often correlate poorly with human judgments and can exhibit spurious length/overlap effects, which may bias query selection.
>
> In our experiments, we explored alternative weak judges, including sentence embedding models and pretrained reward models. However, we did not observe improvements over the k-gram likelihood baseline. Additional results with these alternative weak judges have been added to the appendix. One potential reason is that embedding and reward models can exhibit systematic mismatches with the LLM-as-a-Judge oracle, whereas k-gram likelihoods are relatively more robust to disagreement. Moreover, k-gram models are computationally less costly, making them practical for large-scale query selection, while more complex models would create significant overhead without clear performance gains. As a reference, the embedding model has about 10× longer per-query runtime and the reward model about 100× longer than the k-gram judge.
>
>
> > **Baseline-anchored selection can misidentify the global best**: Because all comparisons are made against a single baseline (a star graph), the model with the highest win-rate versus the baseline need not be the Condorcet or Borda winner overall, especially under intransitive or near-tied preferences. This is a known pitfall in dueling-bandit settings and active ranking when the comparison graph lacks diversity.
>
> We acknowledge that baseline-anchored selection may not always identify the global winner, especially when preferences are intransitive or nearly tied. That said, measuring win-rate against a single baseline LLM is a common practice in LLM-as-a-Judge benchmarks [1,2] and is much more cost-efficient than full pairwise comparisons, which scale quadratically with the number of models. For example, on AlpacaEval, which evaluates 54 models on a pool of 700 queries, a full pairwise comparison would require 700 * 54 * 53 = 2,529,400 oracle annotations. In contrast, baseline-anchored evaluation reduces this to 700 * 53 = 37,100 pairwise comparisons.
> Overall, we utilize the current practice as it allows evaluating many candidate models efficiently and could still accurately identify the best LLM when the selected baseline LLM exhibits reasonable performance.

---

> > ### Author Response · Authors · 2025-11-26
> > **Response to Reviewer NpL6 (2/2)**
> >
> > > **Missing guarantees for the greedy objective**: The method uses an entropy/MI-style greedy policy. Greedy near-optimality holds under adaptive submodularity for certain objectives, but the paper does not show that its objective satisfies these conditions, nor does it provide sample-complexity bounds for identification.
> >
> > Our motivation for the proposed approach builds on the near-optimal guarantees established by Chen et al. [3] for binary symmetric channels. We extend this line of work by reducing the model selection problem to ternary channels. In this work, we introduce the novel problem of model selection for pretrained LLMs and provide a principled and practical solution. We use a sound greedy strategy as an efficient heuristic for information-based objectives and observe strong performance in practice across all benchmarks. A more comprehensive theoretical analysis is an important extension of this work that we intend to pursue next.
> >
> >
> > > The active learning part of the method is query selection, and model choice is produced after annotation. Could the current title "active model selection" be a bit of misleading?
> >
> > We note that “active model selection” is a standard term in the literature [4,5,6,7,8] that refers to the setting where queries are selected actively to identify the best model. In this sense, our use of the term is consistent with prior work.
> >
> > [1] Li, Xuechen, Tianyi Zhang, Yann Dubois, Rohan Taori, Ishaan Gulrajani, Carlos Guestrin, Percy Liang, and Tatsunori B. Hashimoto. "AlpacaEval: An Automatic Evaluator of Instruction-following Models." GitHub, 2023.
> >
> > [2] Li, Tianle, Wei-Lin Chiang, Evan Frick, Lisa Dunlap, Banghua Zhu, Joseph E. Gonzalez, and Ion Stoica. "From Live Data to High-Quality Benchmarks: The Arena-Hard Pipeline." LMSYS blog, 2024.
> >
> > [3] Chen, Yuxin, Hossein Hashani, Amin Karbasi, and Andreas Krause. "Sequential Information Maximization: When Is Greedy Near-Optimal?" COLT, 2015.
> >
> > [4] Zhao, Bin, Fei Wang, Changshui Zhang, and Yangqiu Song. "Active Model Selection for Graph-Based Semi-Supervised Learning." IEEE ICASSP, 2008.
> >
> > [5] Liang, Shen, Yanchun Zhang, and Jiangang Ma. "Active Model Selection for Positive Unlabeled Time Series Classification." IEEE ICDE, 2020.
> >
> > [6] Gardner, Jacob, Gustavo Malkomes, Roman Garnett, Kilian Q. Weinberger, Dennis Barbour, and John P. Cunningham. "Bayesian Active Model Selection with an Application to Automated Audiometry." NeurIPS, 2015.
> >
> > [7] Okanovic, Patrik, Andreas Kirsch, Jannes Kasper, Torsten Hoefler, Andreas Krause, and Nezihe Merve Gürel. "All Models Are Wrong, Some Are Useful: Model Selection with Limited Labels." AISTATS, 2025.
> >
> > [8] Kay, Justin, Grant Van Horn, Subhransu Maji, Daniel Sheldon, and Sara Beery. "Consensus-Driven Active Model Selection." ICCV, 2025.

---

### Author Response · Authors · 2025-12-03
**Concluding Remarks for Rebuttal Phase**

We thank all reviewers for their time and thoughtful feedback throughout the rebuttal period. Reviewers highlighted several strengths of our work, including the practical motivation of efficient model selection under limited annotation budgets (Reviewer NpL6, Reviewer wYka), the principled MI-based formulation (Reviewer wYka), the model-agnostic and scalable design (Reviewer NpL6, Reviewer wYka), and the comprehensive experimental evaluation (Reviewer wYka, Reviewer LzUR).

The reviewers also raised constructive concerns, which we have addressed in detail. In particular:
- Weak judges: We added ablations demonstrating why simple k-gram weak judges are effective compared to more complex alternatives and discussed their computational benefits.
- Two-parameter model of LLM Selector: We further clarified the motivation for this simplified model and its role in keeping the method practical.
- Baseline-anchored comparisons: We explained how baseline-anchored evaluation aligns with standard LLM-as-a-Judge practice and discussed the trade-offs relative to full pairwise comparisons.
- Method clarity and notation: We added missing definitions and derivations in the revised text and appendix, and corrected issues in Algorithm 1.
- Empirical behavior: We provided explanations for dataset-specific trends (e.g., performance at small budgets) and described why LLM Selector is more robust compared to baseline methods.

We appreciate the reviewers’ insights, which helped us further strengthen the clarity and rigor of the paper. Thank you again for your constructive feedback.

---

### Meta-Review · Area_Chair_Z2Cv · 2025-12-29

**Summary:**

LLM Selector proposes an active learning framework to identify the best LLM from a large candidate pool using limited annotations. The method uses mutual information maximization and a k-gram "weak judges" to guide sequential query selection. Experiments across 6 benchmarks with 151 LLMs show up to 59.6% cost reduction.

Presentation too flawed for acceptance: Multiple independent reviewers found clarity issues; requires major revision.
Contribution incremental: Strong empirical system work, but limited algorithmic/theoretical novelty for top-tier venue. While this can be ignored - the challenge is the authors cannot fully explain why/when their method works and in what settings.

**Reviewer Concerns:**

Scores: NpL6 (4/10), wYka (4/10), LzUR (0/10 strong reject) and they would likely remain unchanged. LzUR and authors did not really bridge their disagreements and seemed to complain about each others. It is on the authors to ensure that paper is clear and readable and addresses the concerns raised, justifying their decisions.

**Reviewer Scores:**

Scores: NpL6 (4/10), wYka (4/10), LzUR (0/10 strong reject) and they would likely remain unchanged.

---

### Decision · Program_Chairs · 2026-01-26

Reject